# Victims of Cyberbullying: Feeling Loneliness and Depression among Youth and Adult Chileans during the Pandemic

**DOI:** 10.3390/ijerph19105886

**Published:** 2022-05-12

**Authors:** Jorge J. Varela, Cristóbal Hernández, Rafael Miranda, Christopher P. Barlett, Matías E. Rodríguez-Rivas

**Affiliations:** 1Facultad de Psicología, Universidad del Desarrollo, Santiago 7610658, Chile; matrodriguezr@udd.cl; 2Escuela de Psicología, Universidad Adolfo Ibáñez, Santiago 7941169, Chile; cristobal.hernandez@uai.cl; 3Instituto Milenio para la Investigación en Depresión y Personalidad, MIDAP, Santiago 7820436, Chile; 4Departamento de Psicología, Universidad Continental del Perú, Lima 12001, Peru; rmirandaa@continental.edu.pe; 5Department of Psychological Sciences, Kansas State University, Manhattan, NY 66506-5302, USA; cbarlett@gettysburg.edu

**Keywords:** cyberbullying, depression, loneliness, pandemic

## Abstract

In Chile, during the COVID-19 pandemic, reports of cyberbullying victimization increased for adolescents and younger adults. Research has shown that cyber-victims—adolescents and young adults alike—are at greater risk for mental health problems such as depression as a result of this negative type of aggression. Yet, a paucity of research has examined the individual mechanisms germane to cyber-victim depression. We focused on loneliness for the current study. We hypothesized that cyber-victimization would be positively related to depressive symptoms through increased fears of loneliness and that this effect would differ between adolescents and younger adults. Thus, we examined a sample of 2370 participants from all main regions of Chile aged from 15 to 29 years. Moderated mediation results showed a negative effect of cyberbullying on depression, which was mediated by increased fears of being alone. The effect of frequency of cyberbullying on fear of loneliness was stronger for younger adults compared to adolescents. Our results suggest different mechanisms for both age groups, which can inform prevention programs and their specific activities.

## 1. Introduction

Online communication technologies access and use are widespread across the world. This has opened new opportunities for social interaction and new forms of social relationships between children and adolescents [1], but also for people of all ages and globally. These new technologies have also influenced and expanded the way people communicate. However, at the same time, individuals may create negative online interactions, such as cyberbullying. The purpose of the current study is to expand on the moderated mediation links between cyber-victimization and depression.

### 1.1. Cyberbullying Prevalence

In online contexts, the concern about internet safety has emerged due to, in part, cyberbullying [2,3,4]. Cyberbullying can be defined as its own form of aggression compared to bullying behavior [5] based on unique features such as the use of technologies [6], the hidden identity of the aggressor [7], unlimited boundaries beyond the schools and face-to-face communities [8], a larger audience as a bystander [9], and the power imbalance may not necessarily apply [10]. Indeed, a recent international report from Unesco [11] highlighted unique and shared attributes between cyber and traditional bullying, such as power imbalance, an absence of victim’s response, repetition, and all perpetrators not always acting intentionally—coinciding with the aforementioned past work.

The prevalence of cyberbullying varies across the world and populations. For example, the Digital Civility Index reported in 2020 that 10% of people were victims of cyberbullying (bullied online only) based on a sample of 16,051 individuals from 32 countries, aged between 13 and 74 years. In Chile, the last national study was conducted in 2016, in the First National Survey on Poly-Victimization (Primera Encuesta Nacional de Polivictimización, in Spanish), with students from the 6th to 11th grades. This national study found that 69% reported suffering cyberbullying victimization in the last year [12,13], somewhat consistent with studies conducted in Chile in the past (e.g., [14,15]). Recently, the Minister of Education in Chile received a significant increase in reports of people becoming victims of cyberbullying during the COVID-19 global pandemic. In particular, in 2019, 14 out of every 100 complaints corresponded to cyberbullying, which increased to 26 out of every 100 in 2020 [16]. 

Therefore, we hypothesized an increase in becoming a victim of cyberbullying behavior during the pandemic with negative effects for victims, such as depression, and involving different ages. 

### 1.2. Depression as a Negative Consequence for Cyber-Victims

Research has consistently shown that cyber-victimization is related to a higher risk for depression [17,18]. This association is consistent across different developmental stages, such as young adolescents [19], adolescents [20], college students [21,22], younger adults [23], and adults [24]. The negative effect of cyber-victimization on mental health is consistently shown in myriad meta-analyses (e.g., Zhang et al. [25]; Evangelio et al. [26]). For example, Molero et al. [27], based on 13 studies with a final sample of 7348 adolescents, found a correlation between victims of cyberbullying and depression of 0.28—a moderate to large effect size.

Research on cyberbullying victimization during adulthood is still scarce. Jenaro and colleagues’ [24] systematic review based on 90 different studies in the adult population reported that the percentage of victims ranged from 2.38% to 90.86%. Negative effects on victims were also linked with negative mental health and depression, but with variation among the subjects, which highlights the need to explore more about those individual differences and possible underlying mechanisms. Moreover, during the pandemic, we hypothesized emotional problems for victims of different ages. The pandemic has affected people’s mental health, which can lead to severe psychological crises. According to Hawes et al. [28], people who have been in isolation and quarantine for a long-time experience high levels of anxiety, anger and confusion, and stress. The systematic review of Rubin and Wesley [29] pointed out that, due to the pandemic, people have reported severe trauma, depression, emotional stress, fear, and anxiety due to the infection of themselves and relatives impacted by the loss of quality of life. Therefore, it is imperative to study the psychological effects of cyber-victimization on depression during the pandemic. 

Adolescents and youth are especially vulnerable to mental health due to COVID-19 [30]. During the pandemic, schools were closed, and remote education was initiated, which severely limited students’ social interaction with their peers and teachers [31]. Longitudinal studies show that adolescent anxiety and depression have increased since the pandemic compared to a previous period [32,33,34]. Due to this situation, Liang et al. [35] found that students reported a poorer quality of life and academic difficulties during this period. Zhang et al. [36] found in Shandong province that at least 50% of secondary school students reported having some level of depression symptoms, and a third of them had some level of anxiety during the pandemic situation. In Pakistan, [37] noted that there is a high prevalence of anxiety and depression during the quarantine due to COVID-19. Another China-based study by Chen et al. [38] showed that the prevalence of depression and anxiety among adolescents in China increased by 12% and 6%, respectively, compared to the situation before the onset of the pandemic. Likewise, Liu et al. [39] evaluated mental health indicators of more than 5000 high school students in China during the lockdown and after the lockdown period and found, for depression, a prevalence of 17.35% during the lockdown and a decrease to 13.76% after the lockdown; in the case of anxiety, it fell from 10.35 to 6.73%.

For young adults between 18 and 24 years old, during this developmental stage, many types of behavior are developed, which can lead to either normalcy or mental health illness. Depression, anxiety, and stress (DAS) are the most common mental illnesses among young adults [40,41,42]. Due to these particularities, the effect of the pandemic has had a special impact on this group. According to Seçer and Ulaş [43], following the serious physical and medical effects on individuals, the COVID-19 pandemic is likely to have short- and long-term psychosocial consequences, especially for young people. Nowadays, with the psychological problems of young adults, it is possible that the fear and anxiety caused by the pandemic will trigger various anxiety disorders and similar negative outcomes. For example, Knopf [44] states it is known that for adults, the duration of quarantine, infection fears, boredom, frustration, lack of necessary supplies, lack of information, financial loss, and stigma increase the risk of negative psychological outcomes. A Chinese study by Liu et al. [45] confirms that during COVID-19 young adults faced major psychological challenges. At least one-third of young adults reported having clinically elevated levels of depression (43.3%), anxiety (45.4%), and PTSD symptoms (31.8%). Moreover, a UK study confirmed that for anxiety/depression, there was a strong effect for age, contrary to the effect observed for COVID-19-related anxiety, with very high levels of psychological symptoms in the youngest participants [46]. 

### 1.3. Cyberbullying during the Pandemic

Due to COVID-19, people face higher levels of stress, depression, and anxiety leading to behavioral and emotional problems [47,48]. Likewise, measures to prevent contagion cause people to isolate themselves, causing them to be more vulnerable to attacks by their aggressors [49]. Related, cyberbullying generates uncertainty in the victim regarding who their aggressor is; this generates an increase in their anxiety, leading to what is particularly dangerous for young people who have traumatic experiences [50]. In turn, students who have experienced cyberbullying (victims or aggressors) have significantly lower self-esteem compared to those who have little or no experience with it; it should be added that low self-esteem and cyberbullying are significantly correlated [49].

Victims of cyberbullying can feel lonely [51,52,53], which has been found in adolescents [54] and emerging adults [55]. Moreover, recent studies during the pandemic found cyberbullying victimization correlated with a higher degree of loneliness [56]. Age is also an important factor in loneliness; high school students reported higher levels of loneliness compared to elementary school students; in turn, victimization by cyberbullying correlated with a higher degree of loneliness [56]. Additionally, older age and higher incomes were associated with less depression, less general cyberbullying behaviors, and fewer cyberbullying behaviors in Hubei residents [57].

Adolescents are very vulnerable to becoming cyber-victims during the pandemic. For instance, in June 2020, 80% of young people between 17 and 18 years old were cyberbullied, mostly through the internet [49,58]. It should be added that when people have more personal experience with the pandemic or are close to people who have had the disease, their likelihood of cyberbullying others increases [47]. A study conducted in South Africa detected that in young people who were victims of cyberbullying during the pandemic, their emotional health was affected; for example, depression and even suicide attempts were observed [59]. Likewise, another study conducted in Aktobé, Kazakhstan, showed that high school students who are victims of cyberbullying experience strong emotions such as “anger”, “fear”, and “hatred”, but that they do not turn to other people for help due to the existence of a culture in which the victim is blamed [50]. Additionally, another study conducted in China concluded that adolescents who have experienced cyberbullying during the pandemic reported higher levels of loneliness and lower levels of resilience compared to their counterparts who have not experienced this situation [56]. In addition, another study conducted in Jordan on youth between 19 and 28 years old showed that exposure to cyberbullying is a predictor of low self-esteem and that both are correlated [60].

### 1.4. The Current Study

Even though previous studies recognize the effect of cyberbullying victimization on mental health, less attention has been given to recognizing possible mediators in this relationship, such as feeling lonely, especially during the pandemic. In addition, less is known about the differences between adolescents and younger adults for this relationship. Therefore, we hypothesized that the frequency of cyberbullying victimization would be positively related to increases in depressive symptoms through increased fears of loneliness. We also hypothesized that there would be differences in the different parts of the mechanism conditional to the developmental period of the study participants. The aim of this study is to determine the effects of cyberbullying victimization on the mental health of adolescents and younger adults during the pandemic period.

## 2. Materials and Methods

To test for the proposed mechanism of the effect of cyberbullying on depressive symptoms through fear of loneliness, conditional to the developmental period, we conducted a cross-sectional study using self-report data from a representative national sample from Chile.

### 2.1. Participants

We collected a representative sample of adolescents and young adults from Chile using the Stats Knows methodology [61] in December 2020. Participants were adolescents and young adults from Chile, spanning from 15 to 29 years old.

This methodology utilizes publicly available information offline and on the internet, together with information which participants voluntarily provided in previous studies. Based on the Stat Knows algorithm, people reflecting the required diversity were invited to participate in an online survey in a stratified and random way. The sampling procedure was conducted in two stages: first, the probability of being an internet user was accounted for to calculate the probability of being included in the study, and then in the second stage, a stratified random sample was conducted and calibrated by sex, age, and region of residence within the country, as Chile has 16 different regions. The ethical committee approved this study from Universidad del Desarrollo, Chile. In particular, ethical research protocols and guidelines were met, emphasizing the confidentiality of the information and informed consent of their parents or guardians for adolescents under 18 years old.

The final sample consisted of 2370 participants from all main regions of Chile. The number of participants who reported at least one episode of cyberbullying in the last three months was 1213, with an estimated population of 1,314,421, a sampling error of 2.8%, and a confidence level of 95 % for 15–19-year-old individuals who reported cyberbullying at least once.

### 2.2. Measures

#### 2.2.1. Depressive Symptoms

To measure levels of depression, we used the Patient Health Questionnaire-9 (PHQ-9) [62]. This instrument measures both the existence of Depressive Symptoms and their severity. The PHQ-9 is a self-administered assessment with minimal operational requirements and of proven effectiveness. Previous studies in Chile have used the PHQ-9 in the context of primary health care with a high internal consistency based on a Cronbach’s alpha of 0.84 and a sensitivity and specificity of 88% and 92%, respectively [63]. Moreover, PHQ-9 has an optimal cut-off score of 10 to detect the presence of clinically significant depressive symptoms [64]. The PHQ-9 is based on 9 Likert-type questions (1: No day; 4: Almost every day). Higher scores indicate more presence of depressive symptoms. Cronbach’s alpha for the sample was 0.93.

#### 2.2.2. Cyberbullying

To measure cyberbullying victimization, we used the Cyberbullying and Online Aggression Survey [65]. In particular, we used the single item that measures the frequency of being cyberbullied (“How many times have you been virtually bullied in the last three months?”), structured in a Likert-type response style (0: “Never, it did not happen”; 1: “Once or twice”; 2: “Once a week”; 3: “Several times a week”; 4: “Every Day”.). Higher scores indicate more presence of cyberbullying victimization. This item is particularly useful given that it clearly indicates the frequency of cyberbullying on an intuitive scale. 

#### 2.2.3. Fear of Loneliness

We asked our study participants whether they feared loneliness in their current context with a single item with high face validity: “Thinking about the current times of social confinement and quarantines, how often did you feel with fear of loneliness?”, structured in a frequency Likert-type response style (0: “Never”; 1: “Almost never”; 2: “Sometimes”; 3: “Many times”). This item had the advantage of asking about the fear of loneliness in their current context and not as a direct reaction to cyberbullying, thus overcoming potential confounding with the cyberbullying question. 

### 2.3. Statistical Analyses

#### 2.3.1. Descriptives and Comparison of Experience of Cyberbullying

Given that the main interest of this study was first to evaluate whether young adults reported cyberbullying to the same extent as adolescents, we first split the sample into two meaningful groups based on their age: from 15 to 18 years old and from 19 to 29 years old, as they represent the usual age at high school and university and entry to the workforce. As our variables were not normally distributed given their ordinal or skewed nature (as in the case of depressive symptoms), we tested with an independent sample’s Wilcoxon rank-sum test whether young adults reported cyberbullying intensity and depressive symptoms to the same extent as adolescents.

#### 2.3.2. Mediation Mechanism from Cyberbullying to Depressive Symptoms through Fear of Loneliness

We hypothesized that the frequency of cyberbullying victimization would be positively related to increases in depressive symptoms through increased fears of loneliness. We also hypothesized that there would be differences in the different parts of the mechanism conditional to the developmental period of the study participants. Based on this, we tested a moderated-mediation model within the path analysis framework [66]. All analyses were conducted using R’s [67], “lavaan” library [68]. 

For the mediation part of the model, we simultaneously estimated the effect of cyberbullying victimization frequency on depressive symptoms, usually named path “c” or total effect; the effect of cyberbullying victimization frequency on fear of loneliness, usually called path “a”; and the effect of fear of loneliness on depressive symptoms, usually called path “b”. The mediation effect (usually called indirect effect) results from the product of paths “a” and “b”. Given that indirect effects are usually skewed, a 95% percentile bootstrap with 20,000 draws was used to compensate for deviations from normality [69]. If the 95% CI did not cross zero, then we considered the mediation effect statistically significant. To evaluate the fit of our model to the data, we used the proposed criteria by Hu and Bentler [70], namely: CFI > 0.95, TLI > 0.95, RMSEA < 0.06, SRMR < 0.08. As in our sample, the proportion of men and women was unbalanced, we weighted our estimates and fit indexes by an expansion weight [71] calibrated by age, sex, and region of residence with the “lavaan.survey” package for R [72]. To account for the non-normality of the data resulting from ordinal indicators, we used a diagonally weighted least squares estimator (DWLS), as it has proven to be robust for this kind of deviation [73]. Both weighted and unweighted results are then reported. 

To test whether our proposed mechanism operated conditionally to the developmental period of our participants, we used a multi-group approach [66]. We ran the same model simultaneously in both the adolescent and young adult groups to test for a moderation effect of the developmental period in the different parts of the model. First, a completely constrained model was tested to evaluate whether all paths in the equations were equal for each group. If a good fit of the model was found, then our model did not support our moderation hypothesis. In the second stage, we set path “a” to vary between groups and compared our constrained model with a less restricted model via a likelihood-ratio test [66]. If a significant difference in model fit was found for the less restricted model, we assumed that the effect of cyberbullying victimization on fears of loneliness was conditional to the developmental period. In the third stage, we compared whether the fitness of the model increased when path “b” was set to vary between groups, also implying a differential effect of fear of loneliness on depressive symptoms conditional to the developmental stage. We finally tested whether including path “c” as set to vary increased model fit.

Figure 1 describes the proposed mechanism in which the frequency of cyberbullying has an impact on depressive symptoms by increasing the fear of loneliness. The effect of frequency of cyberbullying to fear of loneliness is thought to be conditional to the age group of the participants, reflecting a differential effect through different life stages. Paths “a”, “b” and “c” are numbered, where “1” refers to the group of adolescents, and “2” refers to the group of young adults. 

## 3. Results

### 3.1. Sample Description

A description of our sample in terms of depressive symptoms, age, sex, frequency of cyberbullying in the last three months, and the experience of fear of loneliness is provided in Table 1. We also provide a correlation matrix of our study variables of interest in Table 2.

It is possible to appreciate that there are no statistically significant differences in the mean values of frequency of cyberbullying victimization (*p* = 0.991) between the young adult (Median = 1 MAD = 1.48) and adolescent groups (Median = 1, MAD = 1.48). However, statistically significant differences were found in the mean values of depressive symptoms (*p* < 0.001), where adolescents showed higher (Median = 15, MAD = 8.9) depressive symptoms than young adults (Median = 12, MAD = 7.4).

### 3.2. Moderated Mediation Analyses

Our first model was the restricted model, in which all paths from the mediation model were constrained to be equal between both groups. As it is possible to observe in Table 3, the model did not fit the data well in both unweighted and weighted samples, so differences can be assumed in the parameters between adolescents and young adults. For model comparison, only the unweighted model are reported for parsimony, given that no substantive differences were found in the results for both samples. 

Our second model allowed path “a” to vary between adolescents and adults. Letting this parameter vary between both groups significantly increased the fit of the model to the data, as indicated by the likelihood-ratio test (Chisqdiff (1) = 10.976, *p* = 0.001). However, it still did not fit the data well with a high RMSEA. 

Our third model allowed both paths “a” and “b” to vary between adolescents and adults. Letting the second parameter also vary between groups significantly increased the fit of the model to the data with respect to the model that only allowed variability between groups on path “a” (Chisqdiff (1) = 25.933, *p* < 0.001) and obtained a good fit to the data. As such, the effect of fear of loneliness on depressive symptoms also varied between adolescents and young adults. 

Our fourth model allowed all paths to vary between adolescents and adults. Letting path “c” vary did not significantly increase the fit of the model to the data with respect to the third model (Chisqdiff (1) = 0.003, *p* = 0.956). As such, the effect of cyberbullying victimization frequency on depressive symptoms was not moderated by the developmental stage of the participants when paths “a” and “b” varied between groups. Given these results, our third model will be further interpreted, and its parameters can be found in Table 4 (weighted parameters in brackets).

Model 3 showed a good fit to the data (see Table 3), so it was possible to interpret it further. First, it is possible to observe that there were no substantively significant differences between the unweighted and weighted samples; however, the weighted model provided an implausible value for path “a” in the adolescent group where cyberbullying predicted less fear of loneliness, contradicting the unweighted data. We consequently ran a model including only the subsample of adolescents in the weighted data to evaluate if this was more likely to be an artifact of the weighting procedure. Results indicated a positive relationship between cyberbullying and fears of loneliness, so the weighted negative value was judged as implausible. We thus report only the unweighted model in text. Regarding the central parameters for our model, there was a positive and statistically significant relationship between the frequency of cyberbullying and fear of loneliness (path “a”). However, this effect was moderated by the developmental stage, being almost five times stronger for young adults (b = 0.267, z (6) = 9.205, *p* < 0.001) than for adolescents (b = 0.091, z (6) = 2.435, *p* = 0.15). On the other hand, we also found a positive and statistically significant relationship between fears of loneliness and depressive symptoms, which was moderated by the developmental stage, where adolescents presented a relationship that was almost 1.5 times higher (b = 3.790, z (6) = 19.301, *p* < 0.001) than young adults (b = 2.680, z (6) = 14.630, *p* < 0.001). The effect of the frequency of cyberbullying victimization (path “c”) remained positive and statistically significant when controlled by the mediator (b = 1.405, z (6) = 7.406, *p* < 0.001). All beta values presented here are unstandardized.

The indirect effect was positive and statistically significant in both groups; however, it was stronger in the young adult group (b1= 0.715, 95% CI = 0.563–0.872, z (6) = 9.069, *p* < 0.001), compared with the adolescent group (b1= 0.346, 95% CI = 0.056–0.640, z (6) = 2.322, *p* = 0.020). This would indicate that the effect of the frequency of cyberbullying victimization on depressive symptoms is partly explained by increases in fears of loneliness associated with cyberbullying and that this mechanism is stronger in young adults than in adolescents. The model for adolescents explained 51% of the variability of depressive symptoms, while 44% in the case of young adults. This difference may be explained by the fact that fears of loneliness were a stronger predictor of depressive symptoms in the case of adolescents. All relevant parameters can be seen in Table 4.

## 4. Discussion

Our study highlights the negative effects on cyberbullying victims during the pandemic for adolescents and younger adults, consistent with previous studies. Moreover, the frequency of cyberbullying increases the fear of loneliness as a reaction to it, and additionally, its effect as a mechanism is higher for younger adults compared with adolescents. In the context of the pandemic and in our increasingly technologically mediated world, these results represent a significant risk factor for mental health, highlighting the need for more mental health support for this population, as many cyberbullying campaigns are aimed mostly toward younger individuals. 

We measured cyberbullying during the pandemic with adolescents and younger adults, hypothesizing similar negative effects for both age groups. As expected, we found no statistically significant differences in the frequency of cyberbullying among adolescents and young adults, which is consistent with previous studies (e.g., Sevcikova and Šmahel [74]). In this case, in the context of the pandemic, we do not see differences between these two groups, which suggests a similar dynamic for both developmental stages. Previous studies have also found no differences between these two age groups. For example, Sevcikova and Šmahel [74] found higher levels of cyberbullying victimization for 1520 adolescents and younger adults compared with older adults in the Czech Republic. Conversely, Cho [75] found the opposite direction with higher ages and more frequency of cyberbullying victimization among adolescents, university students, and working adults in South Korea. 

Regarding the prevalence of depressive symptoms, the adolescent group reports higher levels compared to the younger adult population. However, previous studies on the symptomatology of mental health problems in a pandemic context stated that younger adolescents experienced greater symptomatology compared to adolescents [28,76,77]. Our study provides evidence of cyberbullying victimization effects during the context of the pandemic. In particular, our findings highlight the negative effects of cyberbullying on mental health. However, we did not ask about traditional bullying for adolescents and younger adults because schools, universities, and many workplaces were working remotely because of COVID-19. Considering the significant overlap between these two forms of aggression [15,78,79], future studies can examine the long-term effect on this population in Chile and, eventually, predict more levels of cyberbullying in a face-to-face context. 

Loneliness and cyberbullying are connected, as mentioned by Lowry et al. [80]. Indeed, the use of technological devices such as mobile phones and computers could generate a lonely environment that can lead to aggressive behavior—including cyberbullying. Our results showed that loneliness was a mediator for cyberbullying victimization on depression. Interestedly, other studies have found that loneliness did not mediate the relationship between cyber-victimization and mental health issues (suicidal ideation). Yet, this study took place before the pandemic and included face-to-face victimization in its modeling [81]. We can explain these results based on the negative context generated by the pandemic among adolescents and younger adults for their mental health. During the pandemic, people were more isolated following the national lockdown policies to prevent more contagious diseases, with more negative mental health effects. 

The relationship between loneliness and cyberbullying victimization has had mixed results. We examined the effect of cyberbullying on fear of loneliness, but other studies test the opposite direction. For instance, Brewer and Kerslake [82] examined loneliness as a predictor of cyberbullying victimization with no significant results. Conversely, Şahin [83] found loneliness to be a significant predictor of cyberbullying victimization. Yet, both studies used adolescent samples, and less is known about the older population, such as young adults. Moreover, the underlying mechanisms between cyberbullying victimization and depression during the pandemic need more attention.

Our study found a negative effect of cyberbullying on depression using cross-sectional analysis, but we can hypothesize a long-term negative effect on victims. For example, a recent study examined the longitudinal effects of cyberbullying as a victim or as a perpetrator [18]. As expected, victims of cyberbullying developed negative mental consequences such as internalizing problems, anxiety, and depression, especially in the COVID-19 context [84]. These results highlight the complexity of cyberbullying, especially over time, which can inform prevention programs for this population.

Previous studies have also recognized protective factors for victims of cyberbullying mental health, such as individual and contextual factors [9], for example, self-efficacy for defending himself/herself [41], parental warmth [85], family support [86,87], and school belonging [52]. More studies are needed to explore other possible protective factors for cyberbullying victimization across the lifespan. Protective factors are crucial to building more positive virtual spaces. An interesting finding is that our proposed mediating mechanism was twice as strong in the young adult group compared with the adolescent group, which may indicate a potential target for interventions to alleviate the negative effect of cyberbullying and foster quality social relationships, which can be considered a protective factor for both mental [88] and physical health [89]. Regarding educational implications, these results can be useful for designing educational programs for cyberbullying prevention; as cited in Tanrikulu [90], educational programs that promote social integration and cooperation among students are relevant in preventing this type of violence; in this sense, it is important to involve parents and teachers in these programs, as indicated by Wolfer et al. [91] and Palladino et al. [92].

Our study has some limitations that should be considered. First, we used cross-sectional data, which can limit our results by assuming causality. Therefore, we must interpret our results with caution. Even though we used this type of data, our sample is diverse across the whole country, which helps us to interpret the results in the Chilean context. Second, we used a single question to measure fears of loneliness, which may not capture its multidimensionality [93], and a short questionnaire to measure depression. We recognize other types of information are needed to diagnose depression, such as clinical interviews. Yet, considering the pandemic and the possibility of collecting national data, a short version of the instrument must be used. Moreover, PhQ-9 is a well-developed and validated measure for examining depression [64]. Third, we only focus on two age groups for the current study, hypothesizing more frequency of cyberbullying. Yet, future studies can include other groups considering the use of social media and the internet is widespread across the lifespan. Lastly, we used online self-report data to collect our data, which can provide more access to larger samples due to the lower cost of implementation. Therefore, other versions of the study can complement data collection using face-to-face interviews, including other measures. Finally, we compared different groups to make inferences of conditional effects across the lifespan, which is not equivalent to a longitudinal study.

Despite the previous limitations, our study contributes to the cyberbullying literature by recognizing the effect of depression. Additionally, the effect of the frequency of cyberbullying on depressive symptoms is mediated by increased fears of being alone. Moreover, age explains different results for adolescents compared to younger adults and evaluates how harmful this behavior could be, as mentioned by Celuch et al. [94], leading to even more aggressive online attacks Oksanen et al. [95]. These results may be useful to highlight the different dynamics of the impact of cyberbullying on mental health across the lifespan. 

## 5. Conclusions

Our study evidences the negative effects of cyberbullying across the lifespan during the pandemic, consistent with our hypothesis, reinforcing the need for mental health support for different age groups. In particular, there is a direct effect on depressive symptoms, increasing the risk for other risky behaviors for adolescents and younger adults.

Cyberbullying victimization has negative consequences for mental health across ages. In Chile, during the pandemic, there were increasing reports of this behavior, which is connected with mental health. Thus, the need for prevention programs in the school and other contexts is significant. 

The effect of cyberbullying victimization can also be explained through the fear of feeling alone as an underline mechanism significant for both age groups. This would indicate that the effect of the frequency of cyberbullying victimization on depressive symptoms is partly explained by increases in fears of loneliness associated with cyberbullying. Moreover, we found that this mechanism is stronger in young adults than in adolescents, which can inform prevention programs by providing different components for each age group.

## Figures and Tables

**Figure 1 ijerph-19-05886-f001:**
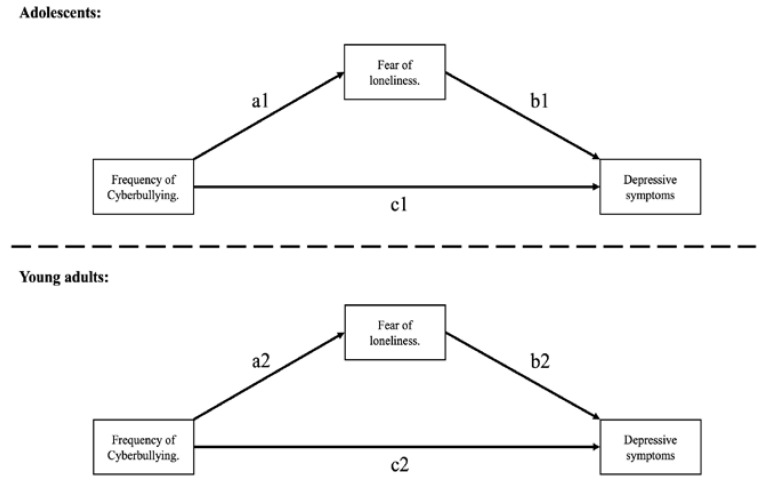
Proposed moderated mediation model.

**Table 1 ijerph-19-05886-t001:** Descriptive statistics.

*Sample Descriptives*				
	Mean (SD)	Value (Categorical)	N	Percentage
Depr. sympt.	14.16 (7.43)	Sub-threshold	837	35.32%
		Clinically significant symptoms	1533	64.68%
Age	20.23 (4.28)			
Sex		Male	433	18.27%
		Female	1937	81.73%
CyberB frequency		Never	1157	48.81%
		Once or twice	822	34.68%
		Once a week	110	4.64%
		Several times/week	267	11.27%
		Every day	14	0.59%
Fear of loneliness		Never	653	27.55%
		Almost never	425	17.93%
		Sometimes	592	24.98%
		Many times	700	29.54%

*Note*. Depr. sympt.—depressive symptoms; age—age, sex—biological sex; CyberB frequency—frequency of cyberbullying; fear of loneliness—frequency of fear of loneliness.

**Table 2 ijerph-19-05886-t002:** Model variables correlation matrix.

Variable	1	2	3
1. Depr. Sympt.	--		
2. CyberBullying	0.27 **	--	
3. Fear of Lone.	0.51 **	0.18 **	--

*Note*. ** indicates *p* < 0.01. Depr. sympt—depressive symptomatology; cyberbullying—cyberbullying frequency; fear of lone.—frequency of fear of loneliness.

**Table 3 ijerph-19-05886-t003:** Fit indexes for path analysis models.

	x2	CFI	TLI	RMSEA	RMSR
Constrained model					
Unweighted	<0.001	0.978	0.956	0.082 [90% CI = 0.055–0.112]	0.033
Weighted	<0.001	0.977	0.953	0.140 [90% CI = 0.113–0.168]	0.064
Path “a” difference model					
Unweighted	<0.001	0.986	0.957	0.081 [90% CI = 0.049–0.118]	0.026
Weighted	<0.001	0.990	0.970	0.111 [90% CI = 0.079–0.147]	0.031
Paths “a” and “b” difference model					
Unweighted	0.971	1.000	1.006	0.000 [90% CI = 0.000–0.000]	0.000
Weighted	0.437	1.000	1.001	0.000 [90% CI = 0.000–0.070]	0.006

*Note*. All models were estimated using a DWLS estimator. Robust versions of the fit indexes are reported. The fit to the data for the model with all parameters free is not reported because the model is saturated.

**Table 4 ijerph-19-05886-t004:** Multi-group mediation model parameters predicting depressive symptoms.

	Adolescents		Young Adults	
	Estimate (S.E.)	*p*-Value	Estimate (S.E.)	*p*-Value
Depr. Sympt.				
Cyberbullying	1.405 [2.024] (0.190 [0.193])	<0.001 [<0.001]	1.405 [2.024] (0.190 [0.193])	<0.001 [<0.001]
Fear of loneliness	3.790 [4.358] (0.196 [0.161])	<0.001 [<0.001]	2.680 [3.085] (0.183 [0.162])	<0.001 [<0.001]
Fear of Loneliness				
Cyberbullying	0.091 [−0.093] (0.037 [0.039])	0.015 [0.017]	0.267 [0.297] (0.029 [0.035])	<0.001 [<0.001]
**Mediation model**				
Indirect effect	0.344 [−0.406] (0.140 [0.174])	0.014 [0.019]	0.715 [0.917] (0.086 [0.102])	<0.001 [<0.001]
Total effect	1.749 [1.618] (0.208 [0.214])	<0.001 [<0.001]	2.120 [2.941] (0.180 [0.200])	<0.001 [<0.001]

*Note*. All beta estimates are not standardized. Depr. sympt.—depressive symptomatology; cyberbullying—frequency of cyberbullying victimization; fear of loneliness—frequency of fear of loneliness. Only the effect of cyberbullying on depressive symptoms was set as equal in both groups. Values in brackets are weighted values, including expansion factors.

## Data Availability

Not applicable.

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
