# Peer review of "Victims of Cyberbullying: Feeling Loneliness and Depression among Youth and Adult Chileans during the Pandemic"

_ijerph, 2022, doi:10.3390/ijerph19105886_

Round 1

Reviewer 1 Report

Overall, I thought the study was well done. My main issue is the writing component. The authors jump from topic to topic from sentence to sentence. For instance, in the abstract: "In Chile, during the pandemic COVID, reports of cyberbullying increased for adolescents 10
and younger adults."....this is one topic. "Victims of cyberbullying are at greater risk for mental health problems such as 11
depression as a result of this negative type of aggression." ....this is another topic. "Previous studies highlight consistent consequences for adolescent and younger adults victims of cyberbullying.".....this is another topic. The abstract sets the stage for the article. If you're discussing COVID then jump to mental health then jump to another topic, it's very confusing for the reader because we're trying to figure out the main theme of the article. Other sections are similar to this. The authors need to finish a topic completely before moving onto another. Stay on topic consistently. There needs to be cohesion consisently throughout the document. If each sentence is a new topic, then there is no cohesion.

Author Response

Thanks for this comment. We edited different sections of the manuscript building more consistent and coherent paragraphs.

Reviewer 2 Report

The work is appropriate and interesting for the scientific community and provides information that may be important for the relevant field of study. However, there are some doubts or comments related to the work:

  • It is recommended to read the latest report related to bullying and cyberbullying by UNESCO. This paper defines bullying and cyberbullying in the light of current research and recommends its use. https://unesdoc.unesco.org/ark:/48223/pf0000374794_spa
  • The sampling error referred to on page 4 is surprising. This is quite a large margin of error considering the sample collected. This is decontextualised and one needs to understand what the total population was to determine it. 
  • Also, in relation to the sample, the large difference between the female and male population is striking. Previous studies have shown that depressive symptomatology and loneliness are problems that are more associated with women. Therefore, with such a difference between men and women, the data may be overestimated. It is not indicated whether this characteristic has been taken into account when constructing the corresponding models.

  • It is also striking that in relation to cyberbullying, only the last three months are asked about. It is not explained whether the questionnaire was passed just after or during the confinement, which would make sense for the last three months. If this is not the case, the answers given are not in line with what we are looking for in the paper.
  • On page 5 it is stated that the questionnaire on fear of loneliness has "high face validity". This should be explained in more detail. 
  • As for the type of analysis carried out, the Student's t-test is indicated but it is not indicated whether the sample has sufficient characteristics to carry out parametric tests. 
  • Although the authors refer to the fact that they have measured fear of being alone as a distinct aspect of cyberbullying, it should not be forgotten that one of the behaviours involved in bullying and cyberbullying is to create an atmosphere of loneliness in the victim. 
  • It is recommended to clearly introduce the clinical and educational implications of the work presented.

Author Response

The work is appropriate and interesting for the scientific community and provides information that may be important for the relevant field of study. However, there are some doubts or comments related to the work:

1. It is recommended to read the latest report related to bullying and cyberbullying by UNESCO. This paper defines bullying and cyberbullying in the light of current research and recommends its use. https://unesdoc.unesco.org/ark:/48223/pf0000374794_spa

Response: Thanks for this comment. We included the latest UNESCO report in the introduction section of the manuscript. 

2. The sampling error referred to on page 4 is surprising. This is quite a large margin of error considering the sample collected. This is decontextualised and one needs to understand what the total population was to determine it. 

Response: We thank the reviewer for pointing this out, as in fact, we made a mistake and considered the sampling error for being threatened on social media so we updated our estimates. The total population of 15 to 29 years considered to be cyberbullyed at least once in Chile was estimated to be 1.314.422. This calculation was made based on the population of 15 to 29 years old living in Chile (4.352.390) and UNESCO’s estimate of the prevalence of cyberbullying in latin-america which is 30.2%. As in our sample we found 1.213 individuals who reported at least one incident in the last three months, we calculated our sampling error to be 2.8% for this particular variable with a 95% of confidence. We updated our manuscript accordingly. 

3. Also, in relation to the sample, the large difference between the female and male population is striking. Previous studies have shown that depressive symptomatology and loneliness are problems that are more associated with women. Therefore, with such a difference between men and women, the data may be overestimated. It is not indicated whether this characteristic has been taken into account when constructing the corresponding models.

Response: We thank the reviewer for this observation. Indeed, it is true that depressive symptomatology and loneliness are problems that are more reported by women than by men, and that our sample was unbalanced. As such, we weighted standard errors and fit indexes by including an expansion weight calibrated by sex, age and region within the country. The expansion weight was included by means of the lavaan.survey package, which was made to include complex sampling structures within the framework of structural equation models in R. Even though no substantive differences were found between both modeling strategies, we now report the weight adjusted version as it is the less biased estimation. 

4. It is also striking that in relation to cyberbullying, only the last three months are asked about. It is not explained whether the questionnaire was passed just after or during the confinement, which would make sense for the last three months. If this is not the case, the answers given are not in line with what we are looking for in the paper.

Response: We thank the reviewer for their comment. Indeed, the data was collected on december 22 to 26th. This corresponds to most of the country in the second phase of our “step by step” plan which included a curfew, and quarantines on the weekends. The second step is just after exiting the total confinement. The plan was localized, so different communes and regions had different regimes. As such, just previously many were or in a total lockdown, or in a partial lockdown. 

5. On page 5 it is stated that the questionnaire on fear of loneliness has "high face validity". This should be explained in more detail. 

Response: We thank the reviewer for highlighting this. Indeed, the question regarding loneliness as presenting a high face validity given that it directly ask the participants to which degree did they experience fears of loneliness in the last three months. Even though we acknowledge that loneliness is a complex and multidimensional construct, this direct and simplified strategy was an acceptable trade-off for us to create a questionnaire that was short enough to collect a nation-wide sample in a short time span. However, we acknowledged this in the limitations section. 

6. As for the type of analysis carried out, the Student's t-test is indicated but it is not indicated whether the sample has sufficient characteristics to carry out parametric tests. 

Response: We thank the reviewer for this observation. Indeed our ordinal responses and depressive symptoms scores were not normally distributed, however no substantive differences were found by using a non-parametric Wilcoxon test. However, we updated our results by including it, as it is the correct way of analyzing this kind of data.  

7. Although the authors refer to the fact that they have measured fear of being alone as a distinct aspect of cyberbullying, it should not be forgotten that one of the behaviours involved in bullying and cyberbullying is to create an atmosphere of loneliness in the victim

Response: We appreciate your comment. Indeed, information on how situations of bullying and cyberbullying focuses on creating an atmosphere of loneliness and isolation in the victims has been added to the text of the article in the discussion section.

8. It is recommended to clearly introduce the clinical and educational implications of the work presented.

Response: Thank you very much for the reviewer comment. A section has been developed that addresses the clinical implications pointed out for mental health and from the education perspective regarding how to work in the school and the family for cyberbullying prevention programs.

Reviewer 3 Report

Thank you for choosing me as a reviewer of this interesting paper.

Research is very important for the period we are experiencing, especially for young people who can suffer from greater emotional fragility in this particular historical moment.
The methodological structure is well taken care of especially in the part concerning the preparation of the survey tool and the analysis methodology. Instead, some doubts arise as regards the selection of the sample as it declares itself to be a random sample but in reality it is a sample 'comparable' to a random sample but is instead postal. In any case, this type of sample generally gives good results and, on the other hand, it is almost never possible to get a real random sample. In addition, this sample is referred to as a simple random one but it is suspected that it is a stratified sample. Can the authors explain this part better?
The paper is an excellent starting point for further study on the psychosocial impact of the pandemic on young people and the authors are invited to continue the observation in the long term as indicated in the conclusions of the paper.

Author Response

  1. Research is very important for the period we are experiencing, especially for young people who can suffer from greater emotional fragility in this particular historical moment. The methodological structure is well taken care of especially in the part concerning the preparation of the survey tool and the analysis methodology. Instead, some doubts arise as regards the selection of the sample as it declares itself to be a random sample but in reality it is a sample 'comparable' to a random sample but is instead postal. In any case, this type of sample generally gives good results and, on the other hand, it is almost never possible to get a real random sample. In addition, this sample is referred to as a simple random one but it is suspected that it is a stratified sample. Can the authors explain this part better?

Response: We thank the reviewer for this observation, as by reading our manuscript again we agree that it is not clearly defined. The sampling was indeed a probabilistic stratified sample conducted in two steps: First there probability to be an internet user was taken into account, and in the second step the sampling strategy was stratified. It was callibrated sex, age and region of residence, as chile is composed by 16 different regions. We further clarified this aspect in the manuscript. 

2. The paper is an excellent starting point for further study on the psychosocial impact of the pandemic on young people and the authors are invited to continue the observation in the long term as indicated in the conclusions of the paper.

Response: Thank you very much for the reviewer comment. Indeed, cyberbullying represents a critical issue during the pandemic whose repercussions will be seen in the short and long term. We will continue researching this topic.

Reviewer 4 Report

The presented article is socially and scientifically relevant. It contributes to the understanding of the factors related to the victimisation and perpetration of cyberhate. It would have been interesting to specify more the relevance of studying cyberbullying and its consequences at the mental level.The objective is clear, but readers like to read the research questions and the hypothesis, it is something more literary. Also, visually, readers tend to search for these paragraphs, adding them.

 The method is statistically rigorous although it would have been interesting to have an overview of the type of schools that agreed to participate, in order to know the % of participants who did not want to take part in the survey. It would also have been relevant to have a more detailed account of the instructions given to the teachers in charge of supervising the survey in the classroom. Furthermore, the evaluator wonders whether the authors focused on cyberbullying or cyberaggression. There is no indication of this. 

I really liked the discussions and the conclusions, could you add some very practical action or direct transfer to society? Some aspects that are easily achievable and useful for teachers, psychologists or social workers. The authors conclude that it would be pertinent to go deeper into the different groups. They could have included more recent work such as that of Celuch and colleagues Celuch, M.; Oksanen, A.; Räsänen, P.; Costello, M.; Blaya, C.; Zych, I.; Llorent, V.J.; Reichelmann, A.; Hawdon, J. Factors associated with online hate acceptance: A cross-national study.
A six-country cross-national study among young adults. Int. J. Environ. Res. Public Health 2022, 19, x. https://doi.org/10.3390; Bauman, Perry, & Wachs; Regarding the triggering effect of terrorist attacks, the work of Oksanen and colleagues may also have documented the debate, e.g. (Oksanen, A., Kaakinen, M., Minkkinen, J., Räsänen, P., Enjolras, B., & Steen-Johnsen, K. (2020). Perceived social fear and cyberhate in the aftermath of the Paris terrorist attacks of November 2015. Terrorism and Political Violence, 32(5), 1047-1066).

The authors should have highlighted the limits of their article from the evaluator's point of view. Finally, English requires revision.
The reviewer suggests some revisions before publication. Finally, congratulations and apologies for taking so long to reply.

Author Response

1. The presented article is socially and scientifically relevant. It contributes to the understanding of the factors related to the victimisation and perpetration of cyberhate. It would have been interesting to specify more the relevance of studying cyberbullying and its consequences at the mental level.

Response: We value this comment. Therefore, we expanded that section of the paper by including for example a recent meta analysis on cyberbullying victim mental health. 

2. The objective is clear, but readers like to read the research questions and the hypothesis, it is something more literary. Also, visually, readers tend to search for these paragraphs, adding them.

Response: We incorporated the hypothesis and the aim of the current study according to the reviewer´s comments.

3. The method is statistically rigorous although it would have been interesting to have an overview of the type of schools that agreed to participate, in order to know the % of participants who did not want to take part in the survey. 

Response: We did not collect data at the school level. Instead, we use social media to collect the sample. Schools and universities were closed during the Pandemic and worked remotely with their students. So, we used an online data collection system described in the method section. We added more information for this section to avoid confusion on this subject.   

4. It would also have been relevant to have a more detailed account of the instructions given to the teachers in charge of supervising the survey in the classroom.

Response: As described above, we did not collect data from schools and universities; therefore, no teachers were involved in the data collection process.

5. Furthermore, the evaluator wonders whether the authors focused on cyberbullying or cyberaggression. There is no indication of this. 

Response: Our study was focused on cyberbullying victimization and the negative consequences on their mental health, such as depression. We examined the manuscript to make sure this information was clear. 

6. I really liked the discussions and the conclusions, could you add some very practical action or direct transfer to society? Some aspects that are easily achievable and useful for teachers, psychologists or social workers. 

Response: Thank you very much for the comment. The practical implications for this part are related to what works preventing cyberbullying based on a systematic review of cyberbullying interventions. For instance, how to work together with teachers and parents to prevent this kind of bullying.

7. The authors conclude that it would be pertinent to go deeper into the different groups. They could have included more recent work such as that of Celuch and colleagues Celuch, M.; Oksanen, A.; Räsänen, P.; Costello, M.; Blaya, C.; Zych, I.; Llorent, V.J.; Reichelmann, A.; Hawdon, J. Factors associated with online hate acceptance: A cross-national study. A six-country cross-national study among young adults. Int. J. Environ. Res. Public Health 2022, 19, x. https://doi.org/10.3390; Bauman, Perry, & Wachs; Regarding the triggering effect of terrorist attacks, the work of Oksanen and colleagues may also have documented the debate, e.g. (Oksanen, A., Kaakinen, M., Minkkinen, J., Räsänen, P., Enjolras, B., & Steen-Johnsen, K. (2020). Perceived social fear and cyberhate in the aftermath of the Paris terrorist attacks of November 2015. Terrorism and Political Violence, 32(5), 1047-1066).

Response: Thank you very much for the suggestions of the papers, they have been very useful to improve the writing of this work.

8. The authors should have highlighted the limits of their article from the evaluator's point of view. 

Response: We have highlighted the limitations of the study according to the reviewer's suggestions.

9. Finally, English requires revision.

Response: Thanks for this comment. We edited the English of the paper. 

Round 2

Reviewer 1 Report

Overall, good. Be sure to proofread and check for grammatical errors including tenses.